# Preparation of Rosin-Based Composite Membranes and Study of Their Dencichine Adsorption Properties

**DOI:** 10.3390/polym14112161

**Published:** 2022-05-26

**Authors:** Long Li, Xiuyu Liu, Lanfu Li, Sentao Wei, Qin Huang

**Affiliations:** 1School of Chemistry and Chemical Engineering, Guangxi Minzu University, Nanning 530006, China; lilong19980227@163.com (L.L.); xiuyu.liu@gxun.edu.cn (X.L.); a17876072393@163.com (L.L.); a1184306866@163.com (S.W.); 2Key Laboratory of Chemistry and Engineering of Forest Products, State Ethnic Affairs Commission, Nanning 530006, China; 3Guangxi Key Laboratory of Chemistry and Engineering of Forest Products, Nanning 530006, China; 4Guangxi Collaborative Innovation Center for Chemistry and Engineering of Forest Products, Guangxi Minzu University, Nanning 530006, China

**Keywords:** rosin-based composite membranes, dencichine, electrostatic spinning technology, notoginseng extracts

## Abstract

In this work, rosin-based composite membranes (RCMs) were developed as selective sorbents for the preparation of dencichine for the first time. The rosin-based polymer microspheres (RPMs) were synthesized using 4-ethylpyridine as a functional monomer and ethylene glycol maleic rosinate acrylate as a crosslinking. RCMs were prepared by spinning the RPMs onto the membranes by electrostatic spinning technology. The optimization of various parameters that affect RCMs was carried out, such as the ratio concentration and voltage intensity of electrospinning membrane. The RCMs were characterized by SEM, TGA and FT-IR. The performances of RCMs were assessed, which included adsorption isotherms, selective recognition and adsorption kinetics. The adsorption of dencichine on RCMs followed pseudo-second-order and adapted Langmuir–Freundlich isotherm model. As for the RCMs, the fast adsorption stage appeared within the first 45 min, and the experimental maximum adsorption capacity was 1.056 mg/g, which is much higher than the previous dencichine adsorbents reported in the literature. The initial decomposition temperature of RCMs is 297 °C, the tensile strength is 2.15 MPa and the elongation at break is 215.1%. The RCMs have good thermal stability and mechanical properties. These results indicated that RCMs are a tremendously promising adsorbent for enriching and purifying dencichine from the notoginseng extracts.

## 1. Introduction

Pharmacologically active natural products have gained unprecedented popularity in recent decades [1]. They have made great contributions historically to drug development, and many of them have had profound effects on our lives. Dencichine (b-N-oxalyl-L-a, b-diaminopropionic acid, b-ODAP), isolated from the roots of panax notoginseng, has a high medicinal value [2]. Dencichine has been reported to show beneficial effects against numerous diseases, such as dispersing stasis and hemostasis, improving platelet number, relieving swelling and pain, kidney diseases, neuroprotection, lowering blood glucose [2,3,4,5] and rheumatic diseases [6]. Natural ingredients in plants are characterized by low concentration [7], the existence of multi-component mixtures, structural diversity and similarity, which pose great challenges to the separation and purification of natural ingredients [8]. Therefore, it is essential to explore efficient extraction and purification processes for dencichine. According to several literature reviews, various methods have been developed to carry ou the separation, determination and enrichment of dencichine in panax notoginseng, such as colorimetry [9], high-performance liquid chromatography (HPLC) [10], gas chromatography–mass spectrometry (GC/MS) [11], liquid chromatographic–tandem mass spectrometric (LC/MS) [12], ultra-high-performance liquid chromatography–tandem mass spectrometry (UPLC-MS/MS) [13], water-methanol method [14] and ultrasonic/microwave-assisted method [15]. The instrumental method employed a specific type of specialization and sophistication of the instruments. The separation method is not highly efficient and profitless to mass production. At the same time, the separated products need to be purified to remove harmful solvents such as acetonitrile. Water-methanol extraction is a traditional and easy-to-operate extraction approach, although inefficient. The ultrasonic/microwave-assisted method in the extraction process has a low separation effect on the structural analogs of dencichine. Molecularly imprinted technology is an easy and efficient method for the preparation of tailor-made polymeric materials with molecular recognition abilities [16]. However, the imprinting of dencichine is problematic and easy to cause template molecule waste. An efficient, low-cost material for dencichine extraction and purification is currently lacking.

In recent years, adsorption-based methods have been widely used in the separation and enrichment of bioactive compounds from many natural products. For example, RPM separation of total alkaloids from coptidis [17] and chitosan membrane purification of artemisinin [18]. The adsorption effect depends on the type and quality of the adsorption material. At present, many kinds of adsorption materials have been developed and deployed for the purification and separation of natural products. The commonly used adsorbents are activated carbon, iron oxide, silica gel, starch, adsorption resin, clay minerals and composite membranes. They use the porous matrices [19,20,21] as an adsorption platform and have good application prospects in the mass transfer and adsorption process. Among them, the composite membranes [22,23,24,25] have excellent thermal stability, high adsorption capacities and stable three-dimensional structure. In addition, they are used repeatedly, so they are widely used for isolating and the enrichment of active ingredients from natural products, such as half terpene [26], flavonoids [27], alkaloids [17] and other active compounds. Poly-ethersulfone (PES) [28], poly-sulfone (PSF) [29], poly-vinylidene fluoride (PVDF) [30,31], poly-vinylpyrrolidone (PVP) [32], poly-acrylonitrile (PAN) [33], poly-vinyl alcohol (PVA) [34] and natural product cellulose [35,36,37,38] are widely used as the polymeric membranes. With people paying more attention to the environment, porous chitosan, cellulose, rosin and other natural polymers appear in people’s field of vision, and the use of natural adsorption materials is becoming more and more prevalent. The rosin-based crosslinking agent has excellent rigidity and contains three double bonds, non-toxic, non-carcinogenic, high abundance, low cost, environmental protection and other advantages. The polymer materials prepared from rosin have the advantages of degradation, high mechanical strength and excellent luster. Rosin-based polymers have been generally implemented not only in traditional fields such as coatings [39] and adhesives [40], but also in emerging fields such as energy [41], environment [42], drug delivery [43] and drug analysis. There are several methods for the preparation of composite membranes, including the casting flow method [44], freeze-drying approach [45] and electrostatic spinning et al. The composite membranes prepared by electrospinning have unique properties such as high specific surface area and uniform nanofiber structure. Electrospinning is currently applied in various applications including electrochemistry, natural product extraction, medicine, the environment and batteries.

In this work, RCMs were developed as excellent adsorbents for the preparation of dencichine for the first time. The preparation of RPMs was carried out using ethylene glycol maleic rosinate acrylate as a crosslinking by precipitation polymerization, and the RCMs were further prepared by electrospinning. Furthermore, we also study the feasibility of RCMs as effective sorptive materials for the dissociation and enrichment of dencichine. The RCMs were characterized by SEM, TGA and FT-IR. Then, the RCMs were evaluated for their sorbent performance of dencichine from notoginseng extracts. The adsorption kinetics of dencichine on RCMs was studied, and the adsorption mechanism was analyzed in detail. Studies have shown that the membranes have excellent application prospects for the extraction of dencichine.

## 2. Materials and Methods

### 2.1. Materials

Dencichine was provided from Chengdu Plant standard pure Biotechnology Co., Ltd. (Chengdu, China). 4-ethylpyridine and 2,2′-azobisisobutyronitrile (AIBN) were provided from Shanghai Aladdin Biochemical Technology Co., Ltd. (Shanghai, China). N, N′-methylenebis (acrylamide) and acetic acid were provided from Shanghai Maclin Biochemical Technology Co., Ltd. (Shanghai, China). Methyl alcohol was obtained from Chengdu Cologne Chemicals Co., Ltd. (Chengdu, China). Glycine-DL-leucine (GL) and glycyl-L-phenylalanine (GP) were provided from Aladdin Chemistry Co. Ltd. (Shanghai, China). HPLC-grade acetonitrile and methanol were provided from Agilent Technologies (Shanghai, China) Co., Ltd. Ethylene glycol maleic rosinate acrylate (EGMRA) was purchased by Wuzhou Sun Shine Forestry & Chemicals Co., Ltd. (Wuzhou, China) [17].

### 2.2. Preparation of RPMs

The functional monomers 4-VP (2.4 mmol) and the EGMEA (0.48 mmol) and MBA (9.6 mmol) were dissolved in 100 mL of methanol in a 250 mL flask. Subsequently, 0.1260 g AIBN was added to the mixed solutions and the organic phase was formed by ultrasound. Then, the mixture was heated followed by mechanical agitation (50 rpm), and heat-polymerized at 70 °C for 11 h. The unreacted monomers were removed by methanol extraction for 48 h. The RPMs were dried under vacuo at 50 °C for 12 h.

### 2.3. Preparation of RCMs

The RCMs were prepared via the electrospinning method. First, the mixed solution with contents of 10 wt% (PAN) and 1 wt% (RPMs) was prepared by adding PAN (2.0000 g) and RPMs (0.2000 g) into DMSO (20 mL) under continuous magnetic stirring at 85 °C for 2.5 h. The mixed solution was extracted 10 mL by using a 10 mL disposable syringe and then the composite membranes were prepared by electrostatic spinning. Electrospinning was carried out at a temperature of 25 ± 5 °C, humidity of 30 ± 5%, the fixed voltage of 20 KV, feeding rate of 0.1 mL/h and a distance of 16 cm. Electrospinning was stopped after 10 h. Finally, RCMs were obtained after methanol extraction for 24 h and vacuum dried at 60 °C for 12 h. The scheme for preparing the RCMs was represented in Figure 1.

### 2.4. Characterization of RCMs and RPMs

The RCMs and RPMs were investigated by FT-IR (MagnA-IR550, Thermo Fisher Scientific, Waltham, MA, USA) in the range 4000–400 cm^−1^. The analysis of the size and morphology of the RCMs and RPMs were performed using field-emission scanning electron microscopy (SEM, Supra 55 Sapphire, Carl Zeiss, Jena, Germany)). Thermogravimetric analysis of the RCMs and RPMs performed using TGA-DSC/DTA analyzer (STA 449 F5, NETZSCH-Gerätebau GmbH, Selb, Germany). The RCMs and RPMs were weighed by analytical balance with an accuracy of 0.1 mg (Practum124-1cn, Sartorius AG, Göttingen, Germany). The zeta potentials of the RCMs and RPMs were measured using a Laser Nanoparticle Size and Zeta Potential Analyzer (Zetasizer Nano, Malvern, UK). The RCM and RPM pore and specific surface area were measured at 77 k using surface area and pore size analyzer (ASAP2020, Micromeritics, Norcross, GA, USA). The mechanical properties of RCM were tested by an electronic universal testing machine (JDL-10000N, Yangzhou Tianfa test Machinery Co., Ltd., Yangzhou, China).

#### 2.4.1. Scanning Electron Microscopy (SEM)

The surface morphology of the RCMs and RPMs was determined by SEM analysis (SEM, Supra 55 Sapphire, Carl Zeiss Germany, Oberkochen, Germany). The samples were evenly coated on the conductive adhesive of the sample sheet and then sprayed with gold for 0.5 h. The surface morphology of the RCMs and RPMs after the samples were sprayed with gold was observed by SEM under low vacuum conditions.

#### 2.4.2. Thermogravimetric Analysis (TGA)

The thermal stability of the RCMs and RPMs was determined by thermogravimetric analysis (TGA) (STA 449 F5, NETZSCH-Gerätebau GmbH, Selb, Germany). The sample was heated from 30 °C to 800 °C for thermal degradation under nitrogen protection at a rate of 10 °C/min.

#### 2.4.3. Dynamic Mechanical Analyzer

The samples were cut to 1 cm in width and 4 cm in length, and the stress–strain curve of the RCMs was measured at a lifting rate of 1 mm/min by the electronic universal testing machine (JDL-10000N, Yangzhou Tianfa test Machinery Co., Ltd., Yangzhou, China). The tensile strength of the RCMs was calculated based on the following Equation (1) [46]:(1)$$\sigma_{b}=\frac{P}{A_{0}}=\frac{P}{\mathrm{bd}}$$
where σ_b_ represents the tensile strength, P represents the maximum tensile load, A represents the cross sectional area of the sample, b represents the width and d represents thickness.

The elongation at the break of RCMs was calculated based on the following Equation (2) [46]:(2)$$\delta =\frac{\Delta L_{b}}{L_{0}}$$
where δ represents the elongation at the break, ΔL_b_ represents the increase in length at the breaking point and L_0_ represents the original length.

### 2.5. HPLC Analysis

All analysis were performed on an Agilent Series 1260 (Agilent Technologies, La Jolla, CA, USA) system, equipped with an autosampler, a quaternary pump, a diode-array detector and a column compartment, controlled by Agilent1260 LC software. Separation was achieved on a ZORBAX SB-C18 analytical column (4.6 × 250 mm, 5 μm, USA) [47]. The mobile phase was 0.05% H_3_PO_4_ aqueous solution and acetonitrile, and the ratio was 95:5. The detection wavelength was 213 nm, the mobile phase flow rate was 1 mL/min and the injected sample volume was 10 µL. The temperature remained at 25 °C.

### 2.6. Static Adsorption

#### 2.6.1. Standard Curve of Dencichine

The content of examined dencichine was determined by HPLC. Dencichine solution with concentration of 0, 0.06, 0.08, 0.10, 0.12, 0.14, 0.16 mg/mL was prepared. The contents of dencichine at different concentrations were analyzed by HPLC (Agilent Technologies, La Jolla, CA, USA) at 213 nm. The standard curve is represented by the following fitting Equation (3).
A = 4071.577C − 0.1569(3)
where A represents the absorption peak area of dencichine solution at 213 nm, C represents the concentration of the different dencichine standards (mg/mL) and the correlation coefficient (R^2^) for this equation was 0.9995.

#### 2.6.2. Static Adsorption of Dencichine on RCMs and RPMs

The RPMs were accurately weighed (0.02 ± 0.0002 g) and placed into round 50 mL conical flasks. After adding 20 mL of dencichine solution, it was adsorbed for 5 h in an 80 rpm air constant temperature oscillator at 25 °C. The content of dencichine after adsorption was determined by HPLC. The adsorption amount of the RPMs on dencichine was calculated based on the following Equation (4) [17].
(4)$$Q_{e}=\frac{\left(C_{0}-C_{e}\right)\times V}{W}$$

Here, Q_e_ represents the amount adsorbed (mg/g), C_e_ represents the equilibrium solution concentration (mg/mL), C_0_ represents the initial concentration (mg/mL), V represents adsorbed solution volume (mL) and W represents the mass of the RPMs (g).

Similarly, the RCMs were accurately weighed (0.15 ± 0.0001 g) and placed into round 50 mL conical flasks. After adding 20 mL of dencichine solution, it was adsorbed for 5 h in an 80 rpm air constant temperature oscillator at 25 °C. The content of dencichine after adsorption was determined by HPLC. The dencichine adsorption amount of the RCMs was calculated based on Equation (4).

#### 2.6.3. Adsorption of Dencichine on Different Mass RCMs and RPMs

The RPMs (10, 15, 20, 25, 30, 35, 40 mg) were weighed with an analytical balance and placed into round 50 mL conical flasks. After adding 20 mL of dencichine solution, it was adsorbed for 5 h in an 80 rpm air constant temperature oscillator at 25 °C. The RCMs (75, 100, 125, 150, 175 mg) were weighed with an analytical balance. After adding 20 mL of dencichine solution, it was adsorbed for 5 h in an 80 rpm air constant temperature oscillator at 25 °C. The content of dencichine after adsorption was determined by HPLC. The dencichine adsorption amount of RCMs and RPMs was calculated based on Equation (4).

#### 2.6.4. Adsorption Kinetics of Dencichine on RCMs and RPMs

The adsorption process of dencichine was studied with the optimal experimental mass. The RCMs and RPMs were used to adsorb dencichine (0.1 mg/mL) in an 80 rpm air constant temperature oscillator at 25 °C. Samples of 0.5 mL were absorbed with 1 mL syringe at 1, 2, 3, 4, 5, 6, 7, 8, 9, 11, 13, 15, 20, 25, 30, 45, 60, 75, 90, 105, 120, 135, 150, 180, 240, 300 min, respectively, and the content of dencichine was determined by HPLC. Subsequently, the adsorption kinetic curve according to the relationship between time and adsorption amount was plotted.

#### 2.6.5. Adsorption Isotherm and Thermodynamics of Dencichine on the RCMs and RPMs

To evaluate the adsorption isotherm of dencichine on RCMs and RPMs, the adsorption was carried out by the addition of  150 mg RCMs or 20mg RPMs into 20 mL of dencichine solution with different concentrations (0.08, 0.10, 0.12, 0.14, 0.16, 0.18 mg/mL). The adsorption equilibrium time was determined by adsorption kinetics experiments. Following the experiment, the adsorption content of RCMs for dencichine in each sample was measured.

To evaluate the thermodynamics of dencichine on the RCMs and RPMs, a series of adsorption tests were carried out, including different temperatures 15, 25, 35, 45 and 55 °C.

#### 2.6.6. Adsorption of Dencichine on the RCMs and RPMs at Different PH

To evaluate the influence of PH on the adsorption performance of dencichine on RCMs and RPMs, a series of adsorption tests were carried out at different pH values (1, 3, 5, 6, 7, 8, 9, 11). The adsorption tests were carried out at 25 °C, 80 rpm with 0.1 mg/mL of dencichine.

### 2.7. Selective Adsorption on the RCMs and RPMs

To evaluate the selective adsorption on the RCMs and RPMs, the adsorption tests were carried out at different solutions (0.10 mg/mL dencichine, 0.10 mg/mL glycine-DL-leucine (GL), 0.10 mg/mL glycyl-L-phenylalanine (GP)). An aqueous GL solution with concentration of 0, 0.06, 0.08, 0.10, 0.12, 0.14, 0.16 mg/mL was prepared. The concentrations of different solutions were measured at 200 nm (GL) using HPLC (Agilent Technologies, La Jolla, CA, USA). The standard curve was represented by the following Equation (5).
A_2_ = 4251.53C_2_ − 0.4393(5)

Here, A_2_ represents the absorption peak area of GL solution at 200 nm, C_2_ represents the concentration of different GL standards (mg/mL) and the correlation coefficient (R2) for this equation was 0.9999.

An aqueous GP solution with a concentration of 0, 0.06, 0.08, 0.10, 0.12, 0.14, 0.16 mg/mL was prepared. The concentrations of different solutions were measured at 210 nm (GP) using HPLC (Agilent Technologies, La Jolla, CA, USA). The standard curve is represented by the following Equation (6).
A_3_ = 7873.23C_3_ − 4.6747(6)

Here, A_3_ represents the absorption peak area of GP solution at 210 nm, C_3_ represents the concentration of different GP standards (mg/mL) and the correlation coefficient (R^2^) for this equation was 0.9999.

## 3. Results and Discussion

### 3.1. Characterization of the RCMs and RPMs

SEM was used to study the surface morphology of RCMs and RPMs, and the results are represented in Figure 2. As can be seen in Figure 2a,b, the RPMs are found to be spherical or nearly spherical objects, and the surfaces of the microspheres had a porous, large surface area. These pores are formed as a result of the diffusion of methanol from the particle to the surface during the polymerization of the polymer. This interconnected porous network provides accessibility and active sites for dencichine adsorption, thus facilitating adsorption. In Figure 2b, it can be seen that the RCMs and the RPMs form a three-dimensional network structure, and the polymers are encased in the membranes. As can be seen from Figure 2c,d, the diameter of the composite membrane fiber prepared by electrostatic spinning is at the nanometer level, with a range of 322.9 ± 73.49 nm. The RCMs present random distributions and are very uniform with dense structure and interconnected large pores. Therefore, electrospinning can provide a rigid frame for the RPMs, which is conducive to the further recycling of the RPMs.

The structure of the RCMs and RPMs was investigated by FT-IR spectroscopy, and the results are represented in Figure 3. In Figure 3a, the RPMs display a stretching vibration peak of -NH occurs at 3308 cm^−1^, a stretching vibration peak of -C=O occurs at 1653 cm^−1^, a -C=C stretching vibration absorption peak is observed at 1521 cm^−1^ and a -COO- stretching vibration absorption peak is observed at 2359 cm^−1^, indicating the successful preparation of RPMs. In the RCM curve, the stretching vibration absorption peak of -C≡N is observed at 2243 cm^−1^, indicating the presence of PAN. Compared with RPMs, the presence of PAN leads to a blue shift in -C=O and a red shift in -C=C. In addition, the FT-IR spectra of the RCMs and RPMs samples are similar, indicating the successful preparation of RCMs.

N_2_ adsorption–desorption experiments were carried out to study the pore volumes [48], pore size distributions, average pore diameters and specific surface areas of the RPMs, and the results are shown in Figure 3b. The pore volume and surface area of the RPMs are 2.756 × 10^−3^ cm^3^/g and 6.1776 ± 0.1204 m^2^/g, respectively. The RPMs have a high specific surface area, which was favorable for the adsorption and extraction of analytes.

TGA curves were presented in Figure 3c, and they were used to describe the thermal stability of the RCMs and RPMs. The RPMs start to decompose at 360 °C, the temperature of the fastest decomposition rate occurs at 381 °C and the maximum decomposition temperature (Tmax) is 462 °C. The decomposition process is divided into two stages, including dehydration in the low temperature zone (100–360 °C) and decomposition in the high temperature zone (381–462 °C). The initial decomposition temperature for the RCMs is 297 °C, the temperature of the fastest decomposition rate occurred at 306 °C, and the Tmax is 333 °C. Decomposition occurs in two stages, including dehydration (100–297 °C) and decomposition (297–333 °C). The second stage of decomposition is due to the breakdown of the PAN in the RCMs. Compared with RCMs, the RPMs have different thermal degradation behavior. The RPMs lost more mass in the high-temperature region from 360 to 500 °C. The TGA results demonstrate that the RCMs and RPMs have excellent thermal stability.

The mechanical properties stability of the membrane long-term stability is one of the significant parameters. Figure 3e shows the stress–strain curves of the as-prepared membranes measured by a dynamic mechanical analyzer. In Figure 3e, it can be seen that the RCMs exhibited excellent mechanical stability with the tensile strength of 2.15 MPa, along with the elongation at the break of 215.1%. The stress–strain curve results demonstrate that the RCMs had excellent mechanical properties.

### 3.2. Optimization Preparation Conditions of the RCMs

The results of the investigation into the preparation conditions of the RCMs are represented in Figure 4. As can be seen in Figure 4a, the adsorption amount of dencichine increases as the RPM concentration increases. When the RPM concentration exceeds 1 wt%, the adsorption amount decreases with the increase in the RPM concentration. The sediment volume is also increased, for a certain PAN concentration, when the polymer concentration increases. Thus, the optimal concentration of the RPMs is 1.0 wt%. In Figure 4b, it can be seen that the adsorption amount of dencichine increases with the increase in the PAN concentration. When the concentration exceeds 10 wt%, the adsorption amount decreases with the increase in the PAN concentration. This may be because the spinning solution with low PAN content has low adhesion, poor spinning effect and poor adsorption amount of the dencichine. PAN content continues to increase, the spinning effect is excellent, and the preparation of the RCMs has also increased the adsorption amount. However, as the content of PAN continues to increase, the spinning solution surface tension increases, the droplet formation of jet flow in the electric field tension is difficult, even blocking the syringe needle which affects the spinning, and the preparation of the morphology of the RCMs will become worse, reducing the adsorption amount. In Figure 4c, it can be seen that the adsorption amount of dencichine increased as the voltage increased. When the concentration exceeds 20 KV, the adsorption amount decreases with the increased voltage. When the voltage concentration is increased, membranes thickness is also increased. With the further increase in the electric field, the drop gradually stays in the electric field for a shorter time and the radius of the RCM center circle decreases. This may be due to the RPMs being wrapped in membranes, thus reduce reducing the adsorption amount.

### 3.3. Adsorption of Dencichine on Different Mass RCMs and RPMs

The adsorption of dencichine (0.1 mg/mL) on different mass RCMs and RPMs is represented in Figure 5. As can be seen in Figure 5a, the adsorption amount of dencichine is increased by increasing the mass of the RPMs. When the mass exceeds 20 mg, the adsorption amount decreases with the increase in the RPM mass. In Figure 5b, it can be seen that the adsorption amount of dencichine increases with the increase in the RCM mass. When the mass exceeds 150 mg, the adsorption amount decreases with the increase in the RCM mass. Under the same concentration conditions, more adsorption sites are provided to dencichine at a small increase, which raises the effective contact area and the amount of dencichine adsorption. With the increase in mass, the RCMs and RPMs had inadequate adsorption of dencichine, and the adsorption amount decreased. In other words, the adsorption of the sorption system stays correlated with the availability of adsorption sites on the surface of the adsorbent and the concentration of the dencichine solution.

### 3.4. Adsorption Kinetics of Dencichine on the RCMs and RPMs

The adsorption kinetics curves of dencichine on the RCMs and RPMs at initial concentrations (0.1 mg/mL) are represented in Figure 6a. The adsorption of dencichine on the RCMs and RPMs showed excellent characteristics of the adsorption kinetics, the adsorption capacity increased with the increase in the adsorption time, and the adsorption rate decreased gradually with increasing adsorption time. As for the RPMs, the fast adsorption stage appeared within the first 15 min, while the slow adsorption stage appeared at 15 to 60 min and the adsorption equilibrium appeared after 150 min. As for the RCMs, the fast adsorption stage appeared within the first 45 min, while the slow adsorption stage appeared at 45 to 90 min and the adsorption equilibrium appeared after 240 min. Compared with RPMs, the RCMs have different adsorption kinetics behavior. This is because dencichine molecules are adsorbed on the 4-VP surface of the RPMs and RCMs during the initial stages. Then, over time, it becomes increasingly difficult for the dencichine to enter the RCMs.

To determine the mass transfer mechanisms and rate controlling, adsorption kinetics of dencichine onto the RCMs and RPMs are evaluated using fitting pseudo-second-order (PSO) and pseudo-first-order (PFO) models [49,50,51,52,53].

PFO adsorption kinetics models:(7)$$\ln\left(Q_{e}-Q_{t}\right)={\mathrm{lnQ}}_{e}-\frac{K_{1}}{2.303}t$$

PSO adsorption kinetics models:(8)$$\frac{t}{Q_{t}}=\frac{1}{K_{2}Q_{e}^{2}}+\frac{1}{Q_{e}}t$$
where Q_e_ represents the adsorbed amount at equilibrium (mg/g), Q_t_ represents the amounts adsorbed at time t (mg/g), K_1_ represents the PFO adsorption kinetics models rate constant (min^−1^) and K_2_ represents the PSO adsorption kinetics models rate constant (g/(mg min)).

The corresponding kinetic parameters calculated by Origin are shown in Table 1. The PFO kinetic model is based on the assumption that adsorption controls diffusion and the PSO kinetic model assumes that the adsorption rate is controlled by the chemisorption process.

Figure 6b shows the relationship between ln (Q_e_ − Q_t_) and time (t), and Figure 6c shows the relationship between t/Q_t_ and time t. It can be seen from the kinetics parameters of both adsorbents presented that the coefficient of determination of PFO kinetics R^2^ (the RPMs) and R^2^ (the RCMs) are 0.9082 and 0.8235, respectively, and that of PSO kinetics R^2^ (the RPMs) and R^2^ (the RCMs) are 0.9999 and 0.9992. The results indicate that the PSO kinetics model fits well with experimental data, and the R^2^ values of the PSO kinetics model are higher than that of the PFO kinetics model. This phenomenon also indicates that chemisorption is a dominant role in the adsorption process.

### 3.5. Adsorption Isotherm and Thermodynamics of Dencichine on the RCMs and RPMs

The adsorption isotherms of dencichine on the RCMs and RPMs at (298 K) with dencichine concentrations of 80, 100, 120, 140, 160 and 180 μg/mL are shown in Figure 7a. As can be seen in Figure 7, the adsorption process of dencichine on RCMs and RPMs was obviously affected by the initial concentration. The dencichine adsorption amount for RCMs and RPMs increased with increasing dencichine concentration.

In order to analyze the adsorption mechanism, fitting Langmuir Freundlich isotherm models to the experimental data (Figure 7b,c) is helpful and allows further understanding of the adsorption mechanism. The equations of these two models are as follows [54,55,56].

Langmuir isotherm equation:(9)$$\frac{1}{Q_{e}}=\frac{1}{Q_{m}}+\frac{1}{K_{3}Q_{m}}\times \frac{1}{C_{e}}$$

Freundlich isotherm equation:(10)$${\mathrm{lnQ}}_{e}={\mathrm{lnk}}_{4}+\frac{1}{n}{\mathrm{lnC}}_{e}$$
where C_e_ represents the concentration of dencichine at equilibrium (mg/mL), Q_e_ represents the dencichine adsorption amount for the RCMs and RPMs at equilibrium (mg/g), 1/n is the dimensionless Freundlich constant, Q_m_ represents the saturation adsorption capacities of monolayer coverage (mg/g), K_3_ represents the Langmuir constant (mL/mg) and K_4_ represent the Freundlich constant (mg/mL).

The Freundlich and Langmuir isotherms for the adsorption of dencichine on the RCMs and RPMs are represented in Figure 7b,c, and the fitting data is shown in Table 2. From the adsorption isotherms data, it is observed that the correlation coefficient (R^2^) for the adsorption of dencichine on the RPMs adsorption has a higher value for the Freundlich equation (0.9976) than the Langmuir (0.9968), indicating that the Freundlich model is more suitable for the adsorption process of dencichine on RPMs. Overall, 0 < 1/n < 1 indicates that the adsorption process easily occurs and has excellent adsorption capacity, 1/n (0.7988) in the Freundlich equation can be seen as a reflection of the easy adsorption behavior. The Langmuir model (0.9570) fits the adsorption data less than the Freundlich model (0.9842) for the adsorption of dencichine on the RCMs. In total, 1/n (2.062) in the Freundlich equation can be seen as a reflection of the adsorption behavior. These results indicated that the adsorption of dencichine on the RCMs was multilayer adsorption. According to the prediction of the Langmuir isothermal model, the maximum adsorption capacity of RPMs for dencichine at 25 °C is 85.11 mg/g. These results further proved the application prospect of RCMs in the separation of dencichine.

To understand the effect of temperature on the adsorption amount of dencichine on the RCMs and RPMs, the results are discussed for the different temperatures, and the results are represented in Figure 8. As shown in this figure, the adsorption process of dencichine on the RCMs and RPMs is affected by temperature. At low temperatures, the dencichine adsorption capacities for the RCMs and RPMs increased with increasing temperature. The contact probability of the RCMs of dencichine increases with the increase in temperature. The RCMs and RPMs have temperature sensitivity because of the hydrogen bonding interaction with the dencichine molecules, and the hydrogen bond is destroyed gradually with the increase in temperature. With the increase in temperature, the adsorption capacity of the RCMs decreased, indicating that high temperature is not conducive to the progress of the adsorption process. This phenomenon proves that chemisorption is dominant in the adsorption process.

### 3.6. Adsorption of Dencichine on the RCMs and RPMs at Different PH Levels

To understand the effect of PH on the adsorption amount of dencichine on the RCMs and RPMs, the results were discussed for the different PH levels, and the results are shown in Figure 9. The adsorption of dencichine on the RCMs and RPMs at the different PH levels of 1, 3, 5, 6, 7, 8, 9 and 11 are shown in Figure 9a.

As shown in this Figure 9a, the maximum dencichine adsorption amount of the RPMs is reached when the pH is 6, and the RCMs are reached when the pH is 7. The results are due to the fact that dencichine is acidic and 4-ethylpyridine is basic, and proves that a weakly acidic or neutral condition could favor the adsorption process. The zeta potentials of the RCMs and RPMs at various pH levels were measured, and the results are presented in Figure 9b. The points of zero charges are at the pH values: 1.7 for RCMs and 8.0 for RPMs, the zeta potential of the RCMs and RPMs decreased as PH increased, and RCMs have more surface negative charge than RPMs. The results indicate that the amino group in PAN reduces the surface charge. Therefore, the adsorption of dencichine by RCMs may be affected by electrostatic interactions. These results further prove that RCMs have exceptional promise for the separation and enrichment of dencichine.

### 3.7. Selective Adsorption on RCMs and RPMs

The selectivity study of RCMs and RPMs was evaluated by using dencichine, two analogues including GL and GP, compared with the adsorption of dencichine in a single solution and the mixed solution (Table 3). Table 3 illustrated the data obtained from the selectivity experiment for both RCMs and RPMs, concerning the adsorption quantity.

In the single solution of three substances, the adsorption amount for dencichine on the RPMs and RCMs is 15.57 mg/g and 1.056 mg/g, respectively. The adsorption amount is better than those for the two analogues. Similarly, in the mixed solution of three substances, the adsorption amount for dencichine on the RPMs and RCMs was 13.79 mg/g and 0.8625 mg/g, respectively, which is significantly higher than those for the two analogues. This phenomenon illustrates a high discrimination property of RCMs and RPMs between dencichine and analogues. These results further proved the application prospect of RCMs in the separation of dencichine.

## 4. Conclusions

In the present work, RPMs were synthesized using 4-VP as a functional monomer and EGMRA as a crosslinker and RCMs were further prepared by electrospinning. Their physicochemical properties and chemical structures were analyzed and characterized. It was observed that RCMs have a negative charge on the surface and have an excellent adsorption effect on dencichine in an aqueous solution and the RCMs showed excellent mechanical and thermodynamic properties. The support of the adsorption isotherm model for Langmuir-Freundlich indicated that the adsorption of dencichine and its analogues by RCMs was multilayers adsorption, and the kinetic adsorption suggested that chemisorption was the main adsorption mechanism. In the meantime, the RCMs showed excellent water solubility and were highly discriminating against dencichine and its analogues, and were conducive to the extraction and purification of a water-soluble bioactive component from natural products. This synthetic method is green, efficient, eco-friendly and cost effective. Therefore, our study provides a novel, efficient and green polymer for the concentration and the purification of dencichine and will open up a new avenue for the purification of dencichine.

## Figures and Tables

**Figure 1 polymers-14-02161-f001:**
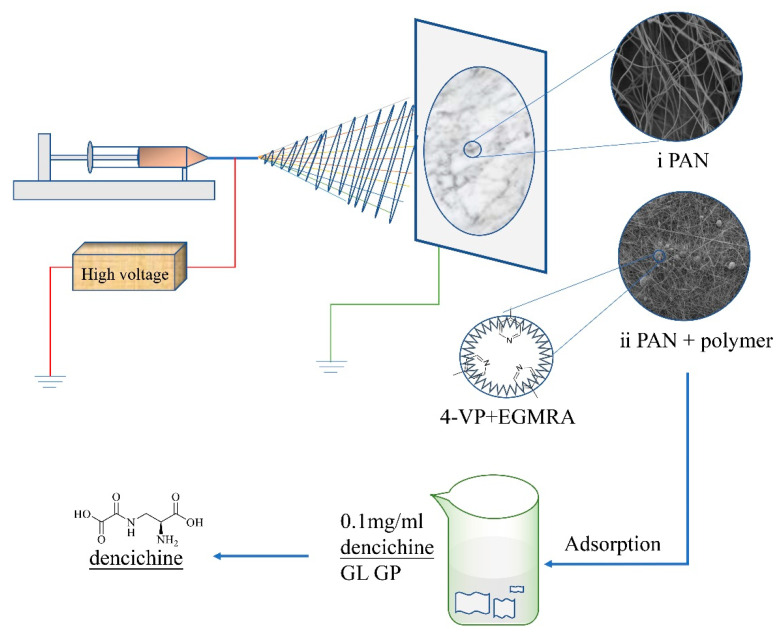
The scheme for preparing the RCMs.

**Figure 2 polymers-14-02161-f002:**
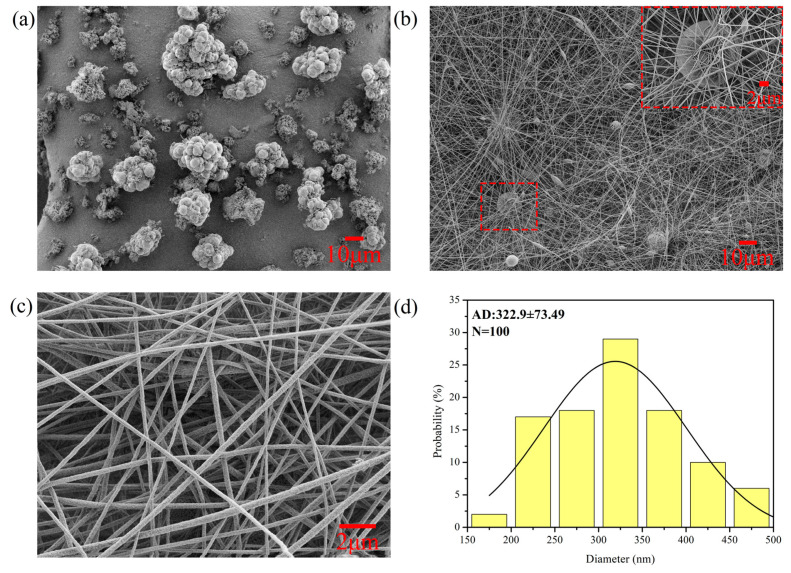
(**a**) SEM images of RPMs; (**b**,**c**) SEM images of RCMs; (**d**) Average particle size and particle size distribution images of RCMs.

**Figure 3 polymers-14-02161-f003:**
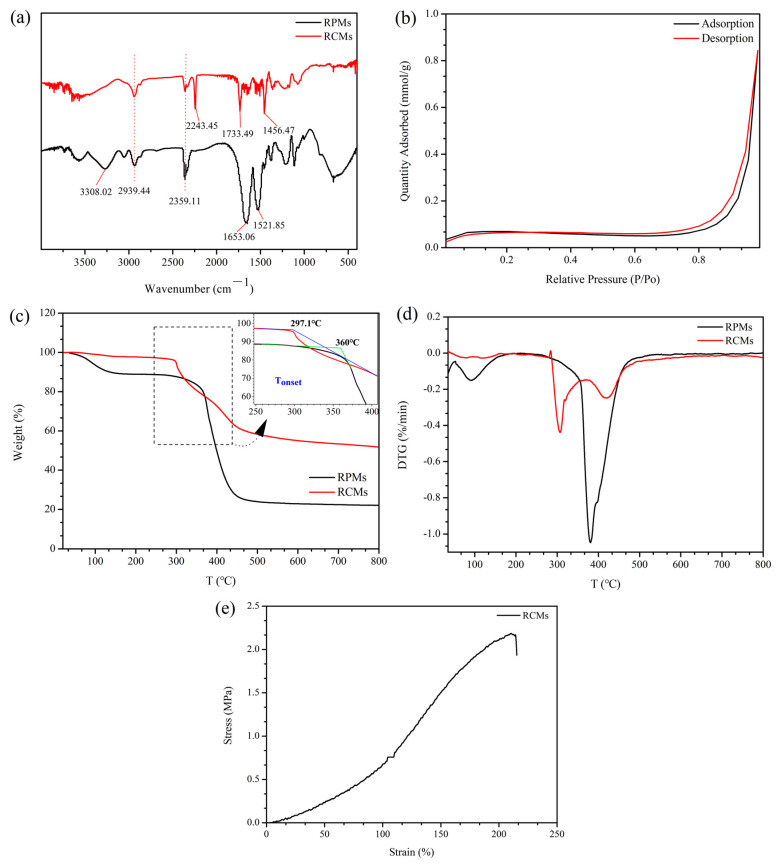
(**a**) FI-IR images of RCMs and RPMs; (**b**) N_2_ adsorption–desorption of RPMs; (**c**) TGA of RCMs and RPMs; (**d**) DTG of RCMs and RPMs (**e**) Stress−strain curves of RCMs.

**Figure 4 polymers-14-02161-f004:**
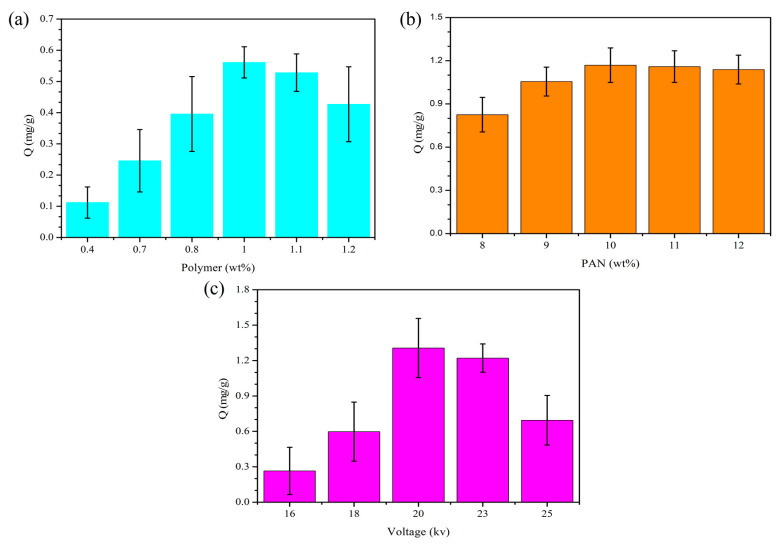
Optimization preparation conditions of the RCMs: (**a**) Polymer content; (**b**) PAN content; (**c**) Voltage adsorption amount.

**Figure 5 polymers-14-02161-f005:**
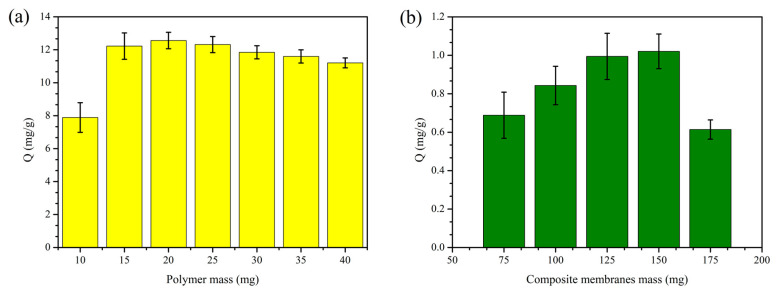
Adsorption of dencichine on different mass RCMs and RPMs; (**a**) RPM adsorption amount; (**b**) RCM adsorption amount.

**Figure 6 polymers-14-02161-f006:**
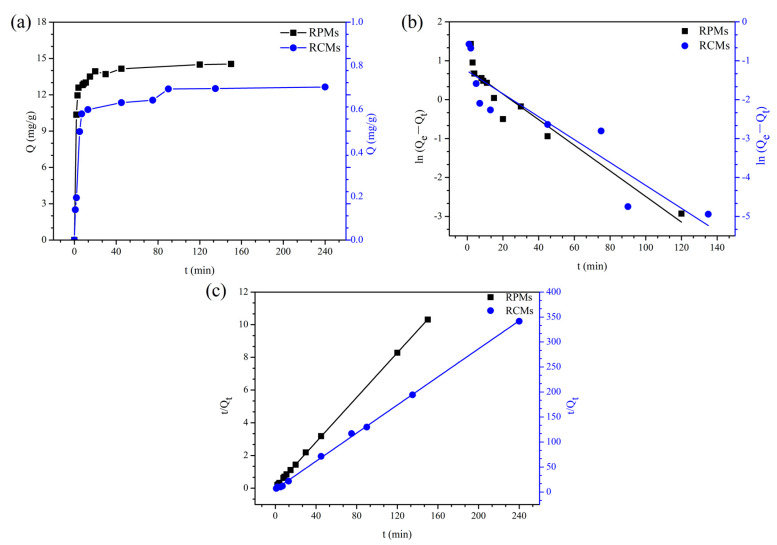
Adsorption kinetics of dencichine on the RCMs and RPMs. (**a**) Adsorption kinetics curves; (**b**) Pseudo-first-order adsorption kinetics model; (**c**) Pseudo-second-order adsorption kinetic.

**Figure 7 polymers-14-02161-f007:**
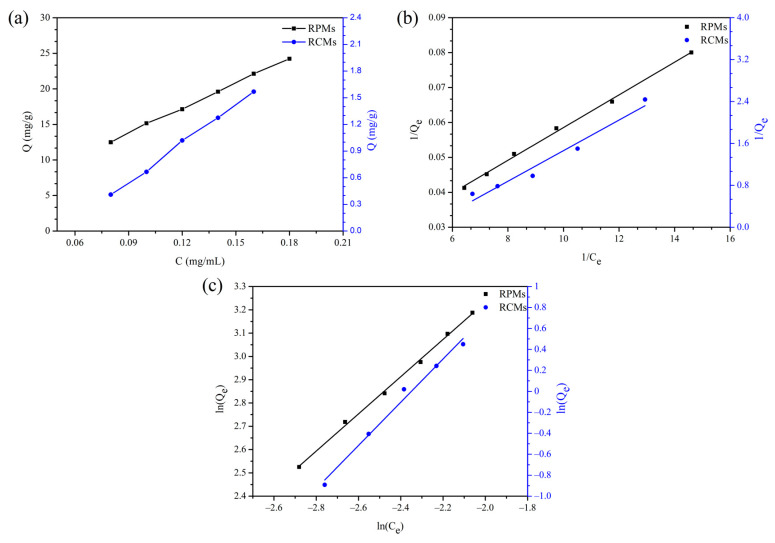
Adsorption thermodynamics of dencichine on RCMs and RPMs. (**a**) Adsorption isotherms; (**b**) Langmuir isotherm model; (**c**) Freundlich isotherm model.

**Figure 8 polymers-14-02161-f008:**
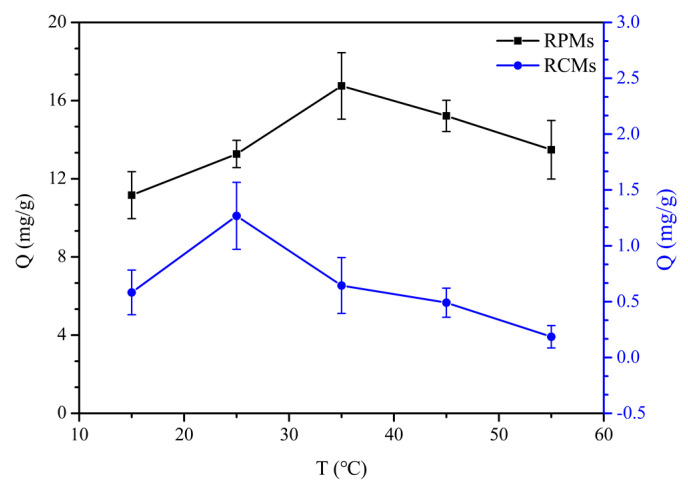
Adsorption thermodynamics of dencichine on RCMs and RPMs.

**Figure 9 polymers-14-02161-f009:**
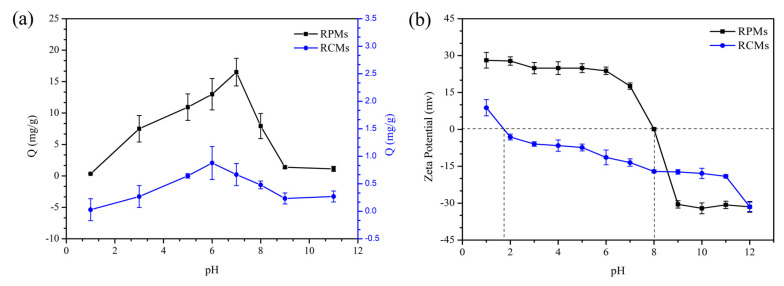
(**a**) Adsorption of dencichine on RCMs and RPMs in different pH conditions; (**b**) The zeta potential of RCM and RPM adsorption amounts.

**Table 1 polymers-14-02161-t001:** Kinetic data of PFO kinetic model and PSO kinetic model.

Samples	PFO Kinetic		PSO Kinetic	
	K_1_ (min^−1^)	R^2^	K_2_ (g mg^−1^ min^−1^)	R^2^
RCMs	0.0678	0.8235	0.3662	0.9992
RPMs	0.0758	0.9082	0.0603	0.9999

**Table 2 polymers-14-02161-t002:** Parameters of Langmuir adsorption model and Freundlich adsorption model.

Samples	Langmuir Isotherm			Freundlich Isotherm		
	K_3_ (mL·mg^−1^)	R^2^	Q_m_ (mg·g^−1^)	K_4_ (mL·mg^−1^)	R^2^	1/n
RCMs	−4.990	0.9570	0.6864	84.19	0.9842	2.062
RPMs	2.511	0.9968	85.11	106.8	0.9976	0.7988

**Table 3 polymers-14-02161-t003:** Preferential adsorption of dencichine on RCMs and RPMs.

Material	The Single Adsorption (mg/g)			The Compound Adsorption (mg/g)		
	Dencichine	GL	GP	Dencichine	GL	GP
RPMs	15.57	1.148	0.5734	13.79	1.261	0.3456
PCMs	1.056	0.0260	0.1841	0.8625	0.2930	0.4494

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
