# Peer review of "Preparation of Rosin-Based Composite Membranes and Study of Their Dencichine Adsorption Properties"

_polymers, 2022, doi:10.3390/polym14112161_

Round 1

Reviewer 1 Report

This paper is a relevant piece of work that provides a very complete characterization of rosin-based composite materials and their performance in adsorbing dencichine, a relevant haemeostatic and neurotoxic agent. The strongest points of the paper are :i) the use of electrospinning to fabricate the composite and their optimization assesing preparation conditions and ii) the analysis of adsorption kinetics where the best fitting models can be correlated to the nature of adsorption.

Despite these two positive points, there are many details that need to be reviewed and that will make the paper more solid and better readable.

1) There are many mistakes at a grammatical level that obscure the understanding of the paper. The paper should be crosschecked by an English native speaker

2) There is a lack of contextualization of adsorption studies at two levels: i) the authors should include and cite previous studies on specific dencichine adsorption and should motivate the choice of their experimental approach for adsorption studies as compared to other techniques that can use porous matrices as adsorption platforms. Cite for instance :

QCM-D: P. Losada-Pérez et al., Engineering the interface between lipip membranes and nanoporous gold: A study by quartz crystal microbalance with dissipation monitoring, Biointerphases 13, 011002 (2018).

EIS: J. McClements et al., Immobilization of Molecularly Imprinted Polymer Nanoparticles onto Surfaces Using Different Strategies: Evaluating the Influence of the Functionalized Interface on the Performance of a Thermal Assay for the Detection of the Cardiac Biomarker Troponin I, ACS App. Mater. Interfaces 13, 27868 (2021).

Heat transfer: G. Wackers et al., Array formatting of the heat-transfer method (HTM) for the detection of small organic molecules by molecularly imprinted polymers, Sensors 14, 11016 (2014).

3) Each instrument used in this work should be included in the Materials Section. For instance, the manufacturer of SEM is not provided nor the analytical balance manufacturer nor precision. No information can be found about the zeta potential analyser.

4) The amount of significant digits provided in Eqs (1) to (4) and in Tables is not acceptable, the authors should correct this equation providing a reasonable number of significant digits according to the right uncertainty.

5) How were the RCMs and RPMs images by SEM taken? They were deposited onto a given surface which is not specified?

6) The panels in the caption of figure 2 are disordered. From figure 2e, the authors should indicate how they calculated the Young modulus and why the value they obtain indicates excellent mechanical properties.

7) The values of the pore volumes obtained by N2 adsorption desorption isotherms should be rewritten with the right number of significant digits

8) Error bars should be added in Figures 3 and 4

9) What do they lines joining points in Figure 6a stand for?

10) Error bars should be added in Figures 7 and 8a

11) What do the authors mean by adsorption ability and how is this calculated? Again, the number of significant digits is way too high

Author Response

Point 1: There are many mistakes at a grammatical level that obscure the understanding of the paper. The paper should be crosschecked by an English native speaker.

Response 1: Sorry for our carelessness. We have made corresponding changes in the revised manuscript.

Point 2: There is a lack of contextualization of adsorption studies at two levels: i) the authors should include and cite previous studies on specific dencichine adsorption and should motivate the choice of their experimental approach for adsorption studies as compared to other techniques that can use porous matrices as adsorption platforms. Cite for instance:

QCM-D: P. Losada-Pérez et al., Engineering the interface between lipip membranes and nanoporous gold: A study by quartz crystal microbalance with dissipation monitoring, Biointerphases 13, 011002 (2018).

EIS: J. McClements et al., Immobilization of Molecularly Imprinted Polymer Nanoparticles onto Surfaces Using Different Strategies: Evaluating the Influence of the Functionalized Interface on the Performance of a Thermal Assay for the Detection of the Cardiac Biomarker Troponin I, ACS App. Mater. Interfaces 13, 27868 (2021).

Heat transfer: G. Wackers et al., Array formatting of the heat-transfer method (HTM) for the detection of small organic molecules by molecularly imprinted polymers, Sensors 14, 11016 (2014).

Response 2: Thank you for your suggestion. We have revised the problem of lack of background in adsorption studies at two levels and cited the following articles:

QCM-D: P. Losada-Pérez et al., Engineering the interface between lipip membranes and nanoporous gold: A study by quartz crystal microbalance with dissipation monitoring, Biointerphases 13, 011002 (2018).

EIS: J. McClements et al., Immobilization of Molecularly Imprinted Polymer Nanoparticles onto Surfaces Using Different Strategies: Evaluating the Influence of the Functionalized Interface on the Performance of a Thermal Assay for the Detection of the Cardiac Biomarker Troponin I, ACS App. Mater. Interfaces 13, 27868 (2021).

Heat transfer: G. Wackers et al., Array formatting of the heat-transfer method (HTM) for the detection of small organic molecules by molecularly imprinted polymers, Sensors 14, 11016 (2014).

Point 3:Each instrument used in this work should be included in the Materials Section. For instance, the manufacturer of SEM is not provided nor the analytical balance manufacturer nor precision. No information can be found about the zeta potential analyser..

Response 3: Sorry for our carelessness. We have revised accordingly.

Point 4: The amount of significant digits provided in Eqs (1) to (4) and in Tables is not acceptable, the authors should correct this equation providing a reasonable number of significant digits according to the right uncertainty.

Response 4: Thank you for your suggestion. We have made corresponding modifications for the significant digits.

Point 5:How were the RCMs and RPMs images by SEM taken? They were deposited onto a given surface which is not specified?.

Response 5: Sorry for our carelessness. The surface morphology of the RCMs and RPMs was determined by SEM analysis ( SEM, Supra 55 Sapphire, Carl Zeiss Germany ). The samples were evenly coated on the conductive adhesive of the sample sheet and then sprayed with gold for 0.5 h. The surface morphology of RCMs and RPMs after the samples were sprayed with gold was observed by SEM under low vacuum conditions. We have revised accordingly in the revised manuscript.

Point 6: The panels in the caption of figure 2 are disordered. From figure 2e, the authors should indicate how they calculated the Young modulus and why the value they obtain indicates excellent mechanical properties.

Response 6: Sorry for our carelessness. Young modulus is Figure 2g and not Figure 2e. We have revised accordingly. The calculation of the Young modulus has been supplemented in this paper. Our experimental study shows that the along with the elongation at the break of RCMs is 215.1%, which can indicate that they have good mechanical properties.

Point 7:The values of the pore volumes obtained by N2 adsorption desorption isotherms should be rewritten with the right number of significant digits.

Response 7: Yes, we have revised accordingly.

Point 8: Error bars should be added in Figures 3 and 4.

Response 8: Yes, we have revised accordingly.

Point 9: What do they lines joining points in Figure 6a stand for?

Response 9: I'm sorry for the reading problem caused by the drawing. The connection points in FIG. 6A is due to the double Y-axis of the drawing, which has no practical significance.

Point 10:Error bars should be added in Figures 7 and 8a.

Response 11: Yes, we have revised accordingly.

Point 11:What do the authors mean by adsorption ability and how is this calculated? Again, the number of significant digits is way too high.

Response 11: I'm very sorry for the difficulty in reading caused by language expression. The adsorption quantity represents adsorption amount, which is calculated by Formula (4). We have revised accordingly.

Reviewer 2 Report

The paper entitled “Preparation of Rosin-based Composite Membranes and Study of Their Dencichine Adsorption Properties “ by long Li, et al. report rosin-based composite membranes (RCMs) developed as the selective sorbents for preparation of dencichine. The study is interesting, and the following comments are raised:

  • Significant of the results should be highlighted in abstract beside adsorption.
  • Reference should support all the used equations in the present study.
  • SEM images, need to provide higher resolution with fiber diameter distribution.
  • In Figure 3, triple sample should be used for statical study with standard deviation., similar in Figure 4.
  • FTIR resulted must compare with previous work in the same text.
  • Several composite membrane materials and applications were recently developed for purification and separation the authors should refer to introduction such as : Journal of Alloys and Compounds , Volume 886, 15 December 2021, 161169, Materials Letters Volume 306, 1 January 2022, 130965, Environmental Nanotechnology, Monitoring & Management Volume 14, December 2020, 100314, MDPI Polymers 2020, 12(11), 2597.

Author Response

Point 1: Significant of the results should be highlighted in abstract beside adsorption.

Response 1: Many thanks for this suggestion. We have revised the description regarding to this issue in the revised manuscript.

Point 2: Reference should support all the used equations in the present study.

Response 2: Thanks for your suggestion. we have revised accordingly.

Point 3: SEM images, need to provide higher resolution with fiber diameter distribution.

Response 3: Many thanks for this suggestion. We have revised it to provide a higher resolution fiber diameter distribution in the paper.

Point 4: In Figure 3, triple sample should be used for statical study with standard deviation., similar in Figure 4.

Response 4: Yes, we have revised accordingly.

Point 5: FTIR resulted must compare with previous work in the same text.

Response 5: Many thanks for this suggestion. We have revised the description regarding to this issue in the revised manuscript.

Point 6: Several composite membrane materials and applications were recently developed for purification and separation the authors should refer to introduction such as : Journal of Alloys and Compounds , Volume 886, 15 December 2021, 161169, Materials Letters Volume 306, 1 January 2022, 130965, Environmental Nanotechnology, Monitoring & Management Volume 14, December 2020, 100314, MDPI Polymers 2020, 12(11), 2597.

Response 6: Thank you for your suggestion. We refer to the following articles in the development of composite membranes: Journal of Alloys and Compounds , Volume 886, 15 December 2021, 161169, Materials Letters Volume 306, 1 January 2022, 130965, Environmental Nanotechnology, Monitoring & Management Volume 14, December 2020, 100314, MDPI Polymers 2020, 12(11), 2597.

Round 2

Reviewer 1 Report

Accept as It is

Reviewer 2 Report

Authors respond to all comments